# Partial Remodeling after Conservative Treatment of Trampoline Fractures in Children

**DOI:** 10.3390/children10020282

**Published:** 2023-02-01

**Authors:** Laura Zaccaria, Enno Stranzinger, Theodoros Xydias, Sabine Schaedelin, Kai Ziebarth, Mike Trück, Vivienne Sommer-Joergensen, Christoph Aufdenblatten, Peter Michael Klimek

**Affiliations:** 1Department of Pediatric Surgery, Hospital Center Biel, 2501 Biel, Switzerland; 2Department of Diagnostic, Interventional and Pediatric Radiology, Inselspital, Bern University Hospital, University of Bern, 3010 Bern, Switzerland; 3Pediatric Radiology, Cantonal Hospital Aarau, 5001 Aarau, Switzerland; 4Department of Clinical Research and Data Analysis, University Hospital Basel, 4031 Basel, Switzerland; 5Department of Pediatric Orthopaedics and Traumatology, University Children’s Hospital Bern, Inselspital Bern, 3010 Bern, Switzerland; 6Department of Pediatric Surgery, Cantonal Hospital Lucerne, 6002 Lucerne, Switzerland; 7Department of Pediatric Surgery, University Children’s Hospital Basel, 4056 Basel, Switzerland; 8Department of Pediatric Orthopaedics and Traumatology, University Children’s Hospital Zurich, 8032 Zurich, Switzerland; 9Department of Pediatric Surgery, Kantonsspital Aarau, 5001 Aarau, Switzerland

**Keywords:** trampoline, trampoline fracture, proximal tibia, positive anterior tilt angle, remodeling

## Abstract

(1) Background: Trampoline fractures (proximal tibia fracture with positive anterior tilt) are increasing. This study represents the first attempt to determine the extent of remodeling in these fractures after conservative treatment (2) Methods: This Swiss prospective multicenter study included children aged 2 to 5 years with a trampoline fracture who were radiologically examined on the day of the accident and after one year. In addition, the anterior tilt angle was compared between the injured and unaffected tibia. Remodeling was defined as complete (final anterior tilt angle ≤ 0°), incomplete (smaller but still >0°), or no remodeling. (3) Results: The mean extent of remodeling was −3.5° (95% CI: −4.29°, −2.66°, *p* < 0.001). Among the 89 children included in the study, 26 (29.2%) showed complete, 63 (70.8%) incomplete, and 17 patients (19.1%) no remodeling. Comparison of the anterior tilt angles between the fractured and healthy tibia showed that the anterior tilt angle on the fractured leg was, on average larger by 2.82° (95% CI: 2.01°, 3.63°; *p* < 0.001). (4) Conclusions: Although the anterior tilt angle decreased during the study period, the majority of patients showed incomplete remodeling. In contrast, children with radiological examinations >1 year after the trauma showed advanced remodeling, suggesting that one year is too short to observe complete remodeling.

## 1. Introduction

The popularity of backyard trampolines in Switzerland has been increasing over the past two decades. Jumping on the trampoline has positive effects on the development of physical strength, coordination, and psychological well-being and is an excellent opportunity to improve the perception of body and space. Unfortunately, we noticed a remarkable increase in trampoline fractures [1,2,3,4,5,6,7]. Among these injuries, fractures of the proximal tibia are particularly common in children under 5 years of age [1,8,9,10]. A recent retrospective study [2] at our institution showed increased anterior tilt angles of the proximal tibia in children who had sustained a trampoline injury compared with tilt angles in an age-matched cohort of healthy children.

Trampoline fractures are typically caused by bouncing on the trampoline, especially when the child jumps with an older or heavier child without parental supervision [1,2,3,5,6]. When two children of unequal weight jump on the trampoline, the jump net follows the heavier child. When the lighter child is in the air, while the heavier one is jumping off, the net hits the leg of the lighter child with full force from below. The fracture pattern results from axial compression of the leg (especially the proximal tibia) and concomitant hyperextension of the knee joint in young children with open growth plates. Other forces, such as torsion, varus, or valgus, are usually not present. However, these fractures are not at risk of progressive valgus deformity after fracture healing, as described by the Cozen phenomenon [3,11,12,13]. Crucial to the diagnosis of a trampoline fracture is the combination of an appropriate medical history, typical symptoms (i.e., pain in the proximal tibia after jumping on a trampoline, refusal to walk), and radiologically, an increased anterior tilt angle at the proximal tibia. Measurement of the anterior tilting of the proximal tibia in the lateral plane is an important, valuable radiological tool for the diagnosis of suspected trampoline fractures, even when no fracture line is evident [2]. Other obvious radiological signs of a trampoline fracture include a buckle, torus, or transverse hairline fracture of the proximal tibial metaphysis or scooping of the tibial tubercle notch. Before radiographic measurement of the positive anterior tilt angle at the tibia, it was difficult to diagnose a trampoline fracture [2], and we must assume that these fractures may have been overlooked in the past. Trampoline fractures are typically treated conservatively [10] in a long leg cast for 3–4 weeks.

However, the long-term effects in terms of functionality and stability of the knee joint after a trampoline fracture remains unclear, and currently, there is nothing in the literature on bone remodeling after trampoline fractures in children.

The aim of this study was to investigate the extent of tibial remodeling after trampoline fracture over the course of at least one year. For this purpose, we documented the anterior tilt angle during follow-up, with a focus on checking for the uplift of the tibial plateau with correction of the anterior tilt angle to values ≤0° (range −2° to +2°).

Based on our observations, we proposed two hypotheses:

**Hypothesis I (main hypothesis).** *In children with a trampoline fracture, the bone might remodel within one year, and the pathologically positive anterior tilt angle might become ≤ 0*.

**Hypothesis II.** *In children with a trampoline fracture, the anterior tilt angle may reach the same angle measured on the healthy leg after conservative therapy in a long leg cast*.

## 2. Materials and Methods

The hospital ethics committees of the five participating national children’s hospitals (Aarau, Basel, Bern, Lucerne, and Zurich) approved this prospective observational multicenter study with a retrospective component. The study was sponsored by the Research Council of the Cantonal Hospital Aarau. One statistician from the clinical trial unit in Basel evaluated the blinded radiological measurements and assessed their statistical relevance. This study was registered at: ClinicalTrials.gov, identifier: NCT04028908.

Parents were informed about the study during normal follow-ups. Participation was voluntary and free of charge for the families. Before study initiation, a specific information pamphlet was created and approved by the ethics committees. The families were required to sign it before participating in the study.

The study included children aged 2 to 5 years who had sustained a fracture of the proximal tibia while jumping on a trampoline and who had radiographs of the injured leg taken on the day of the accident and at least one year after the trauma. Clinical examination included assessment of gait, leg length, and axes such as varus, valgus, and ante- or retro-curvature of the lower leg. For the present study, we prospectively enrolled patients that had sustained a new trampoline fracture in 2016. We also retrospectively enrolled some children (13 of 89 patients) of the same age who had sustained a trampoline fracture between 2011 and 2016 to analyze the remodeling of trampoline fractures even more than one year after trauma. In some cases (72 of 89 patients), we had obtained formal consent to x-ray the opposite healthy tibia (only lateral plane) at least one year after trauma. This was performed exclusively on a voluntary basis with the idea of documenting the uplift of the tibial plateau and comparing the anterior tilt angles between the own injured and healthy leg. All children were treated conservatively in a long leg cast with the knee in full extension or 15–20 degrees of flexion for 3–4 weeks.

The initial and follow-up radiographs were evaluated separately by two specialized pediatric radiologists. Fractures were classified as complete fractures or torus fractures. The anterior tilt angle of the tibia was measured according to our previous study [2] in a lateral radiograph of the lower leg. For this purpose, a tangent was drawn through the proximal and distal ends of the tibial epiphyseal plate. The proximal line of the angle was defined by drawing a tangent between the dorsal and middle points of the physis (the anterior point of the physis can be used if the line also intersects the middle point or two lines of the plate were visible, in which case the lower line was used for measurement). The distal part of the angle was determined by drawing a tangent between the dorsal and ventral epiphyseal plates of the distal physis (Figure 1). Strict or standardized lateral images are not necessary for the measurements of the anterior tilt angles. Measurements were performed with an angle measurement tool on a Sectra AB picture archiving and communication system (PACS IDS7TM, Version 19.3, Teknikringen 20, SE-58330 Linkoping, Sweden).

Before and during the study, we searched the literature for anterior tilt angle values, but to date, no normative pediatric anterior tilt angle values have been defined; therefore, we based our evaluations on results from a previous study [2] and defined a physiological angle as ≤0° (range −2° to +2°). All analyses were performed with R Version 3.6.2 (Vienna, Austria).

### Statistics

Continuous variables are expressed as the arithmetic mean, standard deviation (SD), and categorical variables are expressed as the absolute and relative frequencies. The measurements of the two radiologists were averaged prior to analysis, and their agreement was quantified with the intraclass correlation coefficient (ICC). The ICC is calculated based on an analysis of variance. The ICC is estimated by dividing the variation, which was due to the subject-to-subject difference, through the total variance seen in these data. An ICC of 1 indicates that all differences in the measurements are explainable by differences in the subjects and, therefore, that the method is completely reproducible. The Student’s *t*-test was performed to estimate the extent of remodeling. We defined complete remodeling as an anterior tilt angle ≤0° at the end of the exam. We defined incomplete remodeling as any remodeling that resulted in an anterior tilt angle that was less positive than the initial angle measured after the fracture. Risk factors for incomplete remodeling were assessed with a logistic model.

To evaluate our second hypothesis, we assessed the difference between the healthy and broken legs based on a paired *t*-test. Results are expressed as the odds ratio (OR) and 95% confidence interval (95% CI).

## 3. Results

We identified 93 children, aged 2 to 5 years, that had sustained a trampoline fracture during the period of October 2011 to December 2019 (Figure 2). Four children had to be excluded from the study: two children because their parents refused to participate in the study and two other children because they could not provide a lateral radiograph of the lower leg at the final examination. Therefore, the study included 89 children with an evaluable radiograph of the tibia on the day of the injury and a final examination at least one year (range 11–16 months, median 12 months) after the trauma. In addition, 13/89 patients (14.6%) had a final x-ray taken between 2 and 5 years (range 20–64 months, median 36 months) after the injury. The cohort included 40 males (45%) and 49 (55%) females. The mean age was 3.3 years (SD 1.2). In 72/89 patients (80.9%), we could compare anterior tilt angles between the affected and healthy legs.

**Hypothesis I.** *Anterior tilt angle might show remodeling within one year*.

Among the 89 patients, the mean anterior tilt angle at the time of injury was 5.8° (SD 3.7°), and the average value at the final examination was 2.3° (95% CI: 1.41°, 3.26°, *p* < 0.001; Table 1). Thus, the mean extent of remodeling was −3.5° (95% CI: −4.29°, −2.66°, *p* < 0.001) during the observation period. Overall, 26/89 patients showed complete remodeling with a final anterior tilt angle ≤0°. However, 17/89 patients showed no remodeling at all. In 63/89 children with remodeling, the anterior tilt angle remained positive at the final examination, which indicated incomplete remodeling after the trampoline fracture (Table 2 and Figure 3).

**Hypothesis II.** *Anterior tilt angle might reach the same value as on the healthy leg*.

At least one year after the trauma, 72/89 patients received a radiograph of the healthy leg. The anterior tilt angles at the final examination between the fractured and healthy legs were comparable. However, after remodeling, the anterior tilt angle on the fractured leg remained 2.82° (95% CI: 2.01°, 3.63°, *p* < 0.001; Figure 3) larger than on the healthy leg. The anterior tilt angles measured on the healthy proximal tibia in our study population ranged from −9.4° to +6.2° (mean −0.7336°, Table 3).

In 30/72 patients, the anterior tilt angle was ≥0° at the final examination. In these children, the initial anterior tilt angle was significantly larger than in children with an anterior tilt angle ≤0° at the final exam (7.2° vs. 4.8°, *p* = 0.008; Table 4), despite similar age and sex distributions between groups. Similarly, patients with a large tilt angle on the healthy leg also tended to have a large tilt angle on the fractured leg at the final examination, and thus, they were more often classified as incomplete remodeling (Figure 4).

In our study population, torus fractures were clearly more frequent than complete fractures (Table 2 and Table 4).

We found three significant risk factors associated with a lack of remodeling (final anterior tilt angle >0°, Table 5): male sex (OR: 0.82 compared to females, *p* = 0.0252); large anterior tilt angle on the initial x-ray (*p* < 0.001); and complete fracture (OR torus versus a complete fracture was 1.28, *p* = 0.0193).

None of our patients developed a valgus (Cozen phenomenon) or varus deformity. All patients retained a normal gait and an unaffected leg length.

The average ICC was between 0.68 and 0.82, indicating that the anterior tilt angle measurements were highly congruent between the two radiologists (Table 6).

## 4. Discussion

This is the first study to determine the extent of remodeling in trampoline fractures in children. Our data demonstrated that most fractures showed signs of remodeling on radiographs taken at least one year after the injury. During the study period, we observed an average correction at the tibial plateau of −3.5°. Nevertheless, the anterior tilt angles remained larger (pathologic positive values) than the tilt angles in the healthy legs at the final examination. Due to the values of the anterior tilt angle not reaching negative physiological values or the values of the opposite healthy leg, we concluded that remodeling was largely incomplete (Figure 3 and Table 3).

Our observation time was too short to observe complete remodeling, which was confirmed by the fact that most children who had a final examination > 1 year after the trauma showed, on average, advanced remodeling (most anterior tilt angles < 0°) than children who had a final examination one-year post-trauma (Figure 5).

Summarizing, we can say that remodeling occurs, and the conservative treatment is sufficient. However, we cannot predict how long complete remodeling will take. Therefore, we could not reject either of our two hypotheses.

Unfortunately, we were unable to demonstrate which values of anterior tilt angle predict complete, incomplete, or no remodeling and ultimately determine at which anterior tilt angle the fracture should be treated surgically. Our data suggested that a lack of remodeling was associated with the male sex, a large tilt angle at the initial radiological examination, and a complete fracture (Table 5). Moreover, torus fractures occurred more frequently in our cohort, indicating that fracture severity did not affect remodeling (Table 2 and Table 4).

In our study population, the anterior tilt angles of the proximal tibia of the healthy legs ranged from −9.4° to +6.2° (mean −0.7336°, Table 3). This finding suggests that children of this age might have a physiologically positive anterior tilt angle in the absence of a fracture or have been exposed to repetitive trauma to the proximal tibia while jumping on the trampoline (Table 3 and Table 4). In our previous study, we found chronic repetitive changes such as sclerosis of the metaphysis, growth arrest lines (Harris lines), and widening of the growth plates in children who regularly jump on the trampoline [1]. The latter would at least explain why in our previous work [2], the values of the anterior tilt angle in healthy children who had not suffered any trauma on the trampoline were smaller (3.2°, SD 2.8°). An alternative explanation for the positive anterior tilt angles could be that children with physiologically positive anterior tilt angles, a priori, are more vulnerable or predisposed to trampoline fractures than children with less positive or negative anterior tilt angles. In contrast, the large negative anterior tilt angles indicate that a large normal variation is possible at this young age. In turn, our values reflect the normal ranges of the anterior tilt angles in adults, where the tibial slope averages approximately 10° (SD 3°) [14,15]. Due to the current lack of normative data on proximal tibial tilt angles in children, all these possibilities are speculative. This once again underlines the need to collect more data on normal tilt angles for the child population.

As claimed in the previous study [2], our data indicates that an anterior tilt angle ≥0° does not necessarily indicate a fracture. To diagnose a trampoline fracture with certainty, the clinical presentation and the trauma mechanism must be considered.

All children received conservative treatment for the fractures, and the clinical courses were uneventful. We observed no disturbances in the gait and no angular deformities such as varus, valgus, ante- or retro-curvature of the lower leg during follow-up. In contrast to our work, some previous studies reported growth disturbances related to proximal tibial injuries in children [16,17,18] or angular deformities described by Tuten and Cozen [11,19,20]. These findings suggest that trampoline fracture may be a subtype of proximal tibia fracture that occurs in early childhood and does not affect the growth plate. However, it is still unclear whether persistent ventral tilt of the tibial plateau could lead to altered biomechanics in the knee joint in the future. As shown in previous publications [21], an increase in the tibial slope in teenagers might be a risk factor for ACL (anterior cruciate ligament) injuries, especially in adolescents participating in high-risk activities. This phenomenon is attributed to the fact that increased tibial slope affects the biomechanics of the knee in terms of the anterior translation of the tibia relative to the femur, increasing the load on the ACL and creating a tibial shear force that results in ACL injury [22]. Currently, the future effects of positive tilt angles after trampoline fractures on biomechanics in the knee joint are unknown. In general, we recommend a clinical examination 1–2 years after trauma to avoid overlooking patients who have angular deformities, ante- or retro-curvature, and possibly even no remodeling. Especially in children who do not show remodeling, we recommend clinical and radiological follow-up at the latest before the end of growth in order to intervene in time in case of increasing anterior tilt of the proximal tibia.

This study had some major limitations. First, the low number of patients. Despite our substantial number of cases (*n* = 89 patients), we achieved a power of 0.7, which was below the pre-calculated power of 0.8, for which we needed to include at least 125 patients in the study. Second, a major limitation is the time factor. After at least one year, our data showed mostly incomplete remodeling, suggesting that one year is too short for complete remodeling. Third, the anterior tilt angle was measured indirectly by the growth plate, which does not necessarily represent the tibial slope. The angle values were also not verified by MRI or CT scans. Another limitation was that we only compared injured legs with healthy legs within the same study population. Unfortunately, we did not have data for a large control group or normative data.

## 5. Conclusions

In summary, this study represented the first attempt to understand and show the remodeling of trampoline fractures in children. We could demonstrate that conservative treatment was sufficient and that the post-traumatic course was mainly uneventful. In most of our cases, we could show partial remodeling after one year. However, we cannot determine how long complete remodeling will take. In children who do not show remodeling, we recommend prolonged clinical and possibly radiologic follow-up because the consequences of incomplete remodeling and its effects on biomechanics in the knee joint remain unclear to date.

Our findings should encourage further studies to establish the normal range of anterior tilt angles in the proximal tibias of children to better understand this mechanism in the future.

## Figures and Tables

**Figure 1 children-10-00282-f001:**
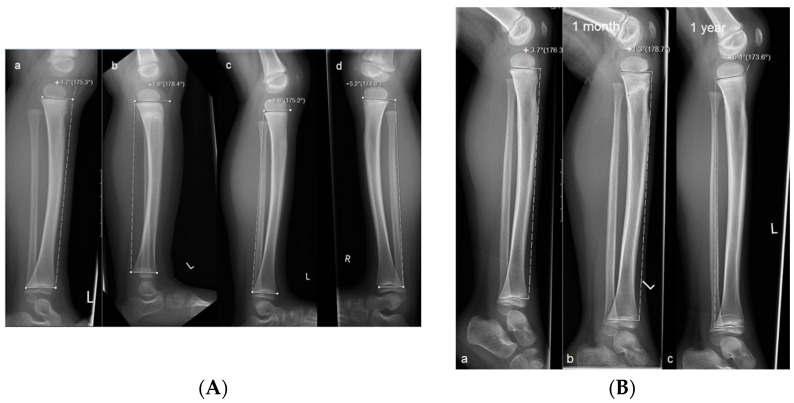
Measurements of the anterior tilt angle of the proximal tibia in children with trampoline fracture. (**A**) Radiographs of a child with a torus fracture. Measurements were taken (a) on the day of the accident; (b) after one month, and (c) after one year; (d) x-ray of the healthy opposite leg. The negative anterior tilt angle (−4.8°) at one year indicates complete remodeling. (**B**) Radiographs of a child with a complete fracture. Measurements were taken (a) on the day of the accident, (b) after one month, and (c) after one year. The negative anterior tilt angle at one year (−6.4°) shows complete remodeling.

**Figure 2 children-10-00282-f002:**
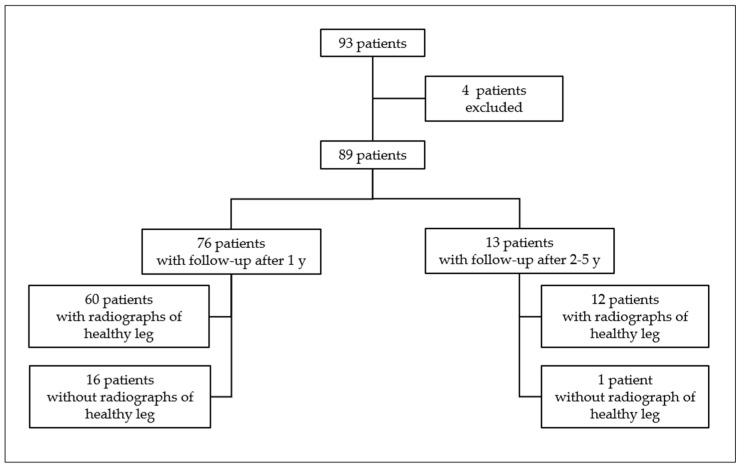
Distribution of our study population.

**Figure 3 children-10-00282-f003:**
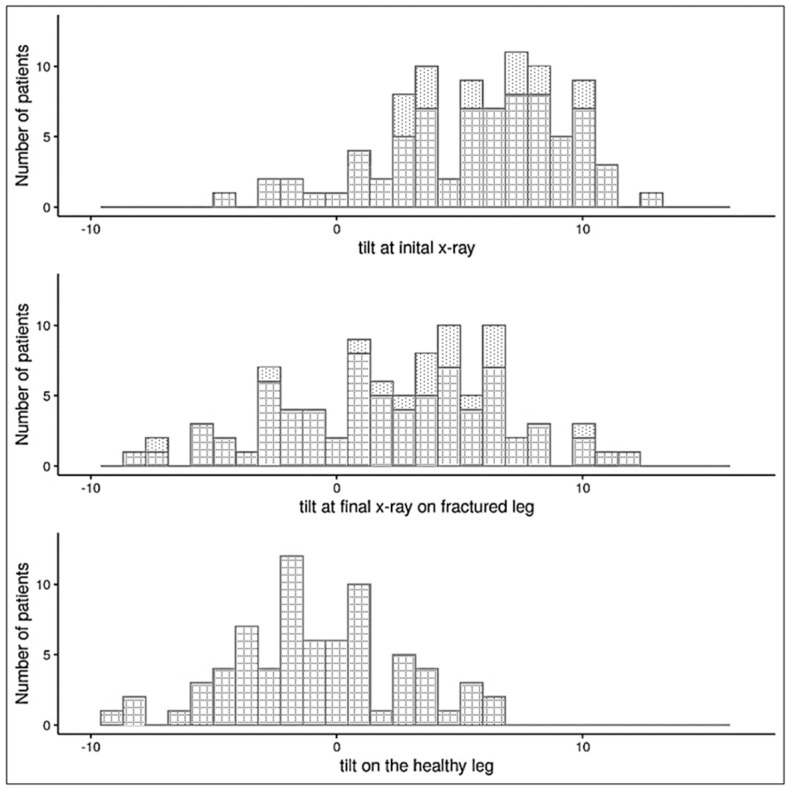
Anterior tilt angles of the proximal tibia in fractured and healthy legs of children with trampoline fractures. The fractured legs were measured in the initial x-ray, taken at the time of the injury, and final x-rays were taken at least one year later. The tilt scale ranges from rather normal (negative) to rather pathological (positive) values. *Dotted bars*: children without an x-ray of the healthy leg (17/89 patients); *checked bars*: children with an x-ray of the healthy leg at the final examination (72/89 patients).

**Figure 4 children-10-00282-f004:**
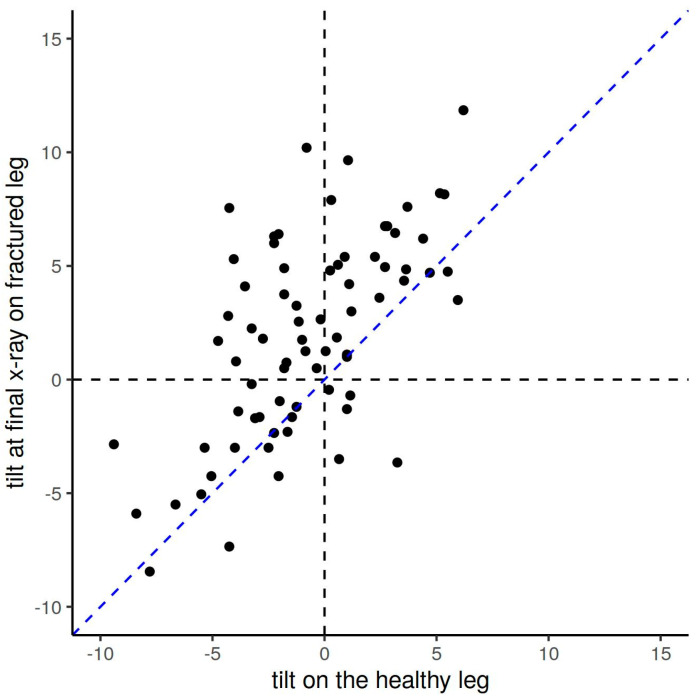
Comparison of the anterior tilt angles between the fractured and the healthy leg at the final x-ray among children with trampoline fractures of the tibia. After remodeling, the anterior tilt angle in the injured leg is still larger than on the opposite healthy leg, as shown here by the majority of anterior tilt angle values (*black dots*) above the bisecting blue line.

**Figure 5 children-10-00282-f005:**
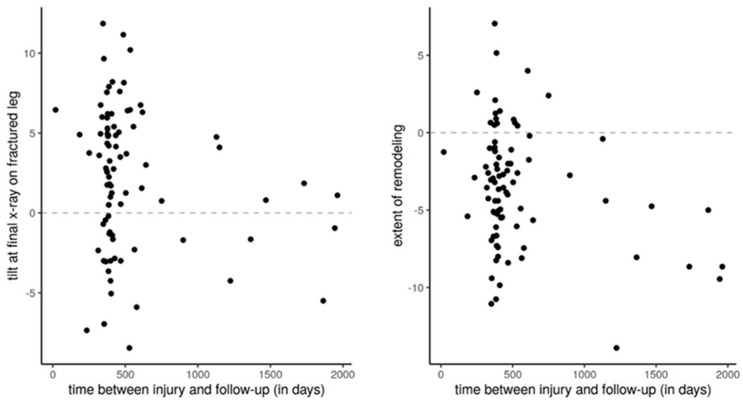
Extent of remodeling over time in children with trampoline fracture of the tibia. Based on the extent of remodeling and the anterior tilt angle at the final examination, we can conclude that patients in whom the final examination was performed >1 year after trauma had better remodeling.

**Table 1 children-10-00282-t001:** Measurements of anterior tilt angles after trampoline fracture in our study population (N = 89) on the day of the injury and one year after trauma.

Time	Anterior Tilt Angle ^a^			
	Min	Max	Mean	Sd
Day of injury	−4.45°	16.60°	5.751°	3.734°
Final exam (fractured leg)	−8.45°	11.85°	2.298°	4.394°

^a^ Values are averages of measurements performed by two radiologists.

**Table 2 children-10-00282-t002:** Characteristics of children with trampoline fractures of the tibia, defined by remodeling (complete/incomplete/no remodeling) and the anterior tilt angle value at the final examination.

Characteristic	Any Remodeling?		Complete Remodeling? Final Anterior Tilt Angle ≤ 0°	
	yes	no	yes	no
*n* (%)	72 (80.9)	17 (19.1)	26 (29.2)	63 (70.8)
Mean age, y (SD)	3.2 (1.1)	3.5 (1.4)	2.8 (1.1)	3.5 (1.2)
male sex, *n* (%)	30 (41.7)	10 (58.8)	7 (26.9)	33 (52.4)
Mean initial tilt, degree (SD)	6.3 (3.6)	3.8 (3.7)	3.5 (3.6)	6.8 (3.4)
Type of fracture, *n* (%) - complete - torus				
	18 (25.0)	4 (23.5)	2 (7.7)	20 (31.7)
	54 (75.0)	13 (76.5)	24 (92.3)	43 (68.3)

**Table 3 children-10-00282-t003:** Summary statistics of anterior tilt angle measurements in children with trampoline fracture of the tibia who received radiographs of the healthy leg on the day of the injury and at least one year after trauma (*n* = 72/89).

Time	Anterior Tilt Angle ^a^			
	Min	Max	Mean	Sd
Day of injury	−4.45°	16.60°	5.6926°	3.853°
Final exam (fractured leg)	−8.45°	11.85°	1.9554°	4.382°
Final exam (healthy leg)	−9.40°	6.20°	−0.7336°	3.453°

^a^ Values are averages of measurements performed by two radiologists.

**Table 4 children-10-00282-t004:** Characteristics of children with trampoline fractures of the tibia also had radiographs of the healthy leg (*n* = 72/89), categorized by the tilt angle measured at the final exam.

Characteristic	Final Tilt > 0°	Final Tilt ≤ 0°
*n* (%)	30 (41.7)	42 (58.3)
Mean age, y (SD)	3.2 (1.1)	3.1 (1.2)
Male sex, *n* (%)	14 (46.7)	18 (42.9)
Mean initial tilt on the fractured leg, degrees (SD)	7.2 (3.7)	4.8 (3.7)
Type of fracture, *n* (%) - complete - torus		
	6 (20.0)	9 (21.4)
	24 (80.0)	33 (78.6)

**Table 5 children-10-00282-t005:** Factors associated with lack of remodeling at one year (anterior tilt angle > 0° at the final exam) after a trampoline fracture of the tibia.

Variable	OR	CI	*p*-Value
Age at final x-ray	0.95	[0.88, 1.02]	0.1885
Male (vs. female)	0.82	[0.70, 0.97]	0.0252
Tilt at initial x-ray	0.96	[0.93, 0.98]	<0.001
Torus (vs. complete fracture)	1.28	[1.05, 1.57]	0.0193

**Table 6 children-10-00282-t006:** Inter-rater agreement between the two radiologists that evaluated radiographs of trampoline fractures of the tibia based on intraclass coefficient (ICC). Only the tilt angle values of patients measured independently by both radiologists were considered.

Time of Measurement	ICC	Patients, *n*	Measurements, *n*
Day of Injury	0.81	89	178
Final x-ray (fractured leg)	0.82	86	172
Final x-ray (healthy leg)	0.68	71	142

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
