# Peer review of "Partial Remodeling after Conservative Treatment of Trampoline Fractures in Children"

_children, 2023, doi:10.3390/children10020282_

Round 1
Reviewer 1 Report
Interesting article on a previously unexplored topic. My comments: Abstract: Please add information that patients after conservative treatment were evaluated. Materials and Methods: Please describe in detail the method of measuring the anterior tilt angle of the tibia. Results: Please give details of the follow-up period of the patients (from to, average). Discussion and Conclusions: Please give one practical conclusion for orthopedic surgeons from Your study.Author Response
Thank you for giving us the opportunity to revise our manuscript. We appreciate your comments and have incorporated them into the manuscript as recommended. We hope that we have addressed the comments to your satisfaction and resolved any concerns.
Reviewer 2 Report
Excellent work, a pioneer in the subject that is addressed. However, I believe that this is a work that should be continued over time. As the authors rightly point out, it has significant limitations, in number, but above all in time. I consider it essential to continue the follow-up and extend the study period to obtain more relevant conclusions for future research.
Author Response
Thank you for considering this manuscript for publication. As recommended by you we will try to continue the follow-up and extend the study period to obtain more relevant conclusions for future research.
Reviewer 3 Report
This manuscript represents a study with the goal to determine the extent of remodelling in trampoline fractures. These are my comments and suggestions:
Abstract:
Nicely and clearly written, no further comment.
Introduction:
Theoretical background is adequate with enough references. Aim of the research is clearly stated.
Methods:
Definition of remodelling should be placed before the Statistics subchapter. Please, state did you check for normality of the data, and how. How did you estimate your sample size?
Results:
I advise to include flow chart diagram of the participants.
Discussion:
If this is the first study exploring remodelling of trampoline fractures, please mention that at the start of the Discussion. I advise to mention the significance of the study - why it is important. Add clinical relevance of the results of the study and suggest further research which could be done in greater detail (altered biomechanics?).
Author Response
Thank you for giving us the opportunity to revise our manuscript. We appreciate your comments and have incorporated them into the manuscript as recommended. We hope that we have addressed the comments to your satisfaction and resolved any concerns. See please the detailed answers below:
Methods:
Definition of remodelling should be placed before the Statistics subchapter
Yes, we agree with them and have changed this as requested.
Please, state did you check for normality of the data, and how
Before and during the work on the study, we searched the literature on anterior tilt angles, trampoline fractures, and remodeling in fractures of the proximal tibia in children, but found no normative pediatric values for the anterior tilt angle. Therefore, we based our assessments on the results of a previous study [2] and defined a physiologic angle of ≤0° (range -2° to +2°).
How did you estimate your sample size?
Despite our substantial number of cases (n=89 patients), we achieved a power of 0.7, which was below the pre-calculated power of 0.8, for which we needed to include at least 125 patients in the study. In the included patients, we quickly found that remodeling did not follow a fixed pattern and was mostly incomplete, so that further radiological controls were not justified for us.
Results:
I advise to include flow chart diagram of the participants
we agree and added a flow chart with the distribution of the study population in the results.
Discussion:
If this is the first study exploring remodelling of trampoline fractures, please mention that at the start of the Discussion
Yes, we agree with them and have changed this as requested.
I advise to mention the significance of the study - why it is important. Add clinical relevance of the results of the study and suggest further research which could be done in greater detail (altered biomechanics?)
Thank you for pointing this out. We have made an effort to highlight the relevant results of the study. You can see the changes in the discussion and conclusion. We now hope to have provided more clarity in this regard and resolved any concerns.